# Metalloprotease-Dependent S2′-Activation Promotes Cell–Cell Fusion and Syncytiation of SARS-CoV-2

**DOI:** 10.3390/v14102094

**Published:** 2022-09-21

**Authors:** James V. Harte, Samantha L. Wakerlin, Andrew J. Lindsay, Justin V. McCarthy, Caroline Coleman-Vaughan

**Affiliations:** 1Signal Transduction Laboratory, School of Biochemistry & Cell Biology and the Analytical and Biological Chemistry Research Facility (ABCRF), University College Cork, Western Gateway Building, T12 XF62 Cork, Ireland; 2Membrane Trafficking & Disease Laboratory, Biosciences Institute, School of Biochemistry & Cell Biology, University College Cork, T12 YT20 Cork, Ireland; 3Department of Biological Sciences, Munster Technological University, T12 P928 Cork, Ireland

**Keywords:** SARS-CoV-2, SARS-CoV-2 spike protein, ACE2, syncytiation, metalloproteases

## Abstract

SARS-CoV-2 cell–cell fusion and syncytiation is an emerging pathomechanism in COVID-19, but the precise factors contributing to the process remain ill-defined. In this study, we show that metalloproteases promote SARS-CoV-2 spike protein-induced syncytiation in the absence of established serine proteases using *in vitro* cell–cell fusion assays. We also show that metalloproteases promote S2′-activation of the SARS-CoV-2 spike protein, and that metalloprotease inhibition significantly reduces the syncytiation of SARS-CoV-2 variants of concern. In the presence of serine proteases, however, metalloprotease inhibition does not reduce spike protein-induced syncytiation and a combination of metalloprotease and serine protease inhibition is necessitated. Moreover, we show that the spike protein induces metalloprotease-dependent ectodomain shedding of the ACE2 receptor and that ACE2 shedding contributes to spike protein-induced syncytiation. These observations suggest a benefit to the incorporation of pharmacological inhibitors of metalloproteases into treatment strategies for patients with COVID-19.

## 1. Introduction

Coronaviruses are a diverse group of positive-sense, single-stranded RNA viruses associated with disease of variable severity in humans [1]. First characterised as agents responsible for mild, self-limiting respiratory infections [2], highly virulent and life-threatening coronaviruses have emerged in recent decades that pose considerable challenges to global health, including the severe acute respiratory syndrome coronavirus (SARS-CoV-1) and the Middle East respiratory syndrome coronavirus (MERS-CoV) [3,4]. The current COVID-19 pandemic is associated with the emergence of the severe acute respiratory syndrome coronavirus 2 (SARS-CoV-2) [5,6,7]. As of August 2022, more than 585 million confirmed cases of SARS-CoV-2-associated COVID-19—including 6.4 million deaths—have been reported to the World Health Organisation [8].

Cellular entry of SARS-CoV-2 is an essential step in its infectious life cycle and is mediated by interaction of the SARS-CoV-2 spike (S) protein with host receptors [9]. The spike protein consists of two functional fragments: The N-terminal S1 subunit possesses a receptor-binding motif that recognises the angiotensin-converting enzyme 2 (ACE2) receptor on the surface of host cells [10,11]; and the C-terminal S2 subunit mediates fusion between the membranes of the virus and the host, which allows the release of the viral genome into the cytoplasmic compartment and its subsequent replication [12,13]. However, similar to other class I viral fusogens, the native SARS-CoV-2 spike protein requires post-translational proteolytic processing at the boundary of the S1 and S2 subunits (S1/S2) by furin and furin-like proteases to prime a fusion-competent state [14]. Following its interaction with the ACE2 receptor, the spike protein is further processed at the S2′ site located immediately downstream of the S1/S2 boundary. This activates the spike protein by exposing a highly conserved segment of hydrophobic residues, known as the fusion peptide [15,16]. The S2′-activation of the SARS-CoV-2 spike protein has been reported to occur via two pathways: an early pathway mediated at the cell surface by the serine protease TMPRSS2 and TMPRSS2-related enzymes [17,18], and a late entry pathway mediated by the cysteine protease cathepsin L within the endolysosomal system [19]. However, cathepsin-dependent entry may represent a cell-culture-specific adaption in the absence of suitable proteases *in vitro*, with clinical coronavirus isolates achieving S2′-activation by serine proteases alone [20]. Therefore, the serine protease TMPRSS2 is considered the predominant activating protease of the SARS-CoV-2 spike protein *in vivo*.

During viral replication within host cells, leakage of the spike protein from intracellular compartments results in the expression of the viral fusogen in the host cell membrane which, upon recognition and interaction with the ACE2 receptor on the surface of neighbouring host cells, facilitates cell–cell fusion and the formation of multinucleated syncytia [21,22]. Although the specific role of virus-induced cell–cell fusion in the natural history of COVID-19 is a matter of debate [23], histopathological evidence suggests that syncytiation is a significant pathomechanistic feature of severe COVID-19 [24,25,26,27]. SARS-CoV-2 can propagate efficiently within syncytia without recognition by the immune system, and dissemination either by direct cell-to-cell transmission [28,29] or dislodgement of syncytial elements [30] facilitates the spread of SARS-CoV-2 within the body. The lymphocytopenia and cytokine storm associated with severe COVID-19 have also been linked to syncytiation, with the cell-in-cell elimination of immunocytes [31] and subsequent pyroptosis [32], contributing to the pathoprogression of the disease.

Recent studies have reported SARS-CoV-2 spike protein-induced cell–cell fusion and syncytiation in the absence of serine proteases [33,34], suggesting that SARS-CoV-2 can also engage host cells lacking the known proteolytic profiles required for viral entry. However, the nature of the ancillary factors necessary to support serine protease-independent fusion remain unclear [35].

Herein, we report a serine protease-independent pathway of SARS-CoV-2 cell–cell fusion and syncytiation, which is dependent on host cell metalloprotease activity. We demonstrate that metalloprotease-dependent SARS-CoV-2 cell–cell fusion is attributable to S2′-activation of the SARS-CoV-2 spike protein, and that metalloprotease inhibitors prevent syncytiation. Furthermore, we demonstrate a reciprocal relationship between the processing of the SARS-CoV-2 spike protein and the ACE2 receptor and implicate for the first time ACE2 proteolysis in SARS-CoV-2 syncytiation. These findings suggest that pharmacological inhibitors of metalloproteases could be repurposed to treat patients with COVID-19.

## 2. Materials and Methods

### 2.1. Reagents and Antibodies

All reagents and materials were purchased from Sigma-Aldrich (Dublin, Ireland) unless otherwise stated. Batimastat (catalogue number: SML0041), camostat mesylate (catalogue number: SML0057) and TAPI-1 (catalogue number: SML0739) were purchased from Sigma-Aldrich. Marimastat was purchased from MedChemExpress (catalogue number: HY-12169). Phorbol 12-myristate 13-acetate (PMA) was purchased from Calbiochem (catalogue number: 524400).

Antibodies used in this study are as follows: 1D4 rhodopsin (monoclonal; 1:1000) was from Invitrogen (catalogue number: MA1-722); FLAG-tag (monoclonal: 1:1000) was from Invitrogen (catalogue number: F3165); B-2 GFP (monoclonal: 1:1000) was from Santa Cruz Biotechnology (catalogue number: sc-9996); human ACE2 C-terminal (polyclonal; 1:1000) was from ProSci Inc. (catalogue number: 3217); human ACE2 N-terminal (polyclonal; 1:2000) was from R&D Systems (catalogue number: AF933-SP); human TMPRSS2 (monoclonal: 1:1000) was from Abcam (catalogue number: EPR3862); SARS-CoV-1 RBD was from Cell Signalling Technologies (catalogue number: 63847S); AC-15 β-actin (monoclonal; 1:10,000) was from Sigma-Aldrich (catalogue number: A5441); FITC-conjugated anti-goat (polyclonal: 1:11,000) was from Jackson ImmunoResearch; HRP-conjugated anti-mouse (polyclonal; 1:5000) and anti-rabbit (polyclonal; 1:5000) secondary antibodies were from Dako Cytomation; infrared secondary antibodies, IRDye™ 800CW goat anti-rabbit IgG (polyclonal; 1:10,000) and IRDye™ 800CW goat anti-mouse IgG (polyclonal; 1:10,000), were from LI-COR Biosciences; IRDye™ 700DX goat anti-mouse IgG (polyclonal; 1:10,000) was from Rockland.

### 2.2. Cell Culture

HEK293T cells (human kidney cells; originally purchased from ATCC, Middlesex, United Kingdom), CaCo-2 cells (human colorectal cells; kindly provided by Dr. John Morgan, University College Cork, Ireland), Calu-3 cells (human lung cells; kindly provided by Prof. John Laffey, National University of Galway, Ireland), and Huh-7 cells (human liver cells; kindly provided by Prof. Liam Fanning, University College Cork, Ireland) were maintained in completed Dulbecco’s Modified Eagle Medium (DMEM) at 37 °C in a humidified atmosphere containing 5% CO_2_. Completed DMEM was prepared by supplementing basal medium with 10% foetal bovine serum, 1% non-essential amino acid solution, 2 mM glutamine, 50 U/mL penicillin, and 50 μg/mL streptomycin. Transfection of plasmids was performed by calcium–phosphate precipitation or Metafectene^®^ Pro (Biontex, München, Germany), as per the manufacturer’s instructions. For transfection, HEK293T, Caco-2, Calu-3, and Huh-7 cells were seeded 24-h prior to the addition of transfection reagent in 6-well plates at a density of 0.5 × 10^6^ cells/well. All cells were routinely tested for mycoplasma species by polymerase chain reaction.

### 2.3. Plasmids

The following plasmids were obtained from Addgene: a plasmid for human ACE2 (pcDNA3.1-hACE2: Addgene plasmid #1786) was a gift from Hyeryun Choe; a plasmid for the ancestral Wuhan-Hu-1 SARS-CoV-2 spike protein (pcDNA3.1-SARS2-Spike: Addgene plasmid #145032) was a gift from Fang Li; plasmids for the SARS-CoV-2 spike proteins of the Delta (pCAGGS-S-B.1.617.2: Addgene plasmid #177097) and Omicron (pCAGGS-S-B.1.1.529: Addgene #180774) variants were gifts from Daniel Conway; a plasmid for human TMPRSS2 (pCSDest-TMPRSS2: Addgene plasmid #53887) was a gift from Roger Reeves; and a plasmid for human ADAM17 (pRK5F-TACE: Addgene plasmid #31713) was a gift from Rik Derynck. Expression plasmids encoding the α-fragment (pCMV-α) and ω-fragment (pCMV-ω) of β-galactosidase were generously gifted by Christian J. Buchholz (Molecular Biotechnology and Gene Therapy, Paul-Ehrlich-Institut, 63225 Langen, Germany) [36]. For generation of the pcDNA3.1-hACE2-L584A construct, site-directed mutagenesis was performed with overlapping QuikChange™-like primer pairs. The mutagenic primers were: 5′-CATTCCGGTGACCGGTTGATGAAACTCGGG-3′ (forward) and 5′-CCCGAGTTTCATCAACTCGTCACCGGAATG-3′ (reverse).

### 2.4. Western Blot Analysis

After the indicated treatments, whole cell extracts were prepared by lysis in radioimmunoprecipitation assay buffer (50 mM Tris-HCl [pH 7.4], 150 mM NaCl, 1 mM EDTA, 1 mM EGTA, 1% (*v*/*v*) Triton X-100, 1% sodium deoxycholate, and 0.1% SDS), freshly supplemented with 1 mM sodium orthovanadate and EDTA-free protease inhibitor mixture (cOmplete™, Roche). Extracts were clarified by centrifugation (16,000× *g*, 10 min, 4 °C), and the protein concentration in the supernatant was quantified using a bicinchoninic acid assay (Pierce™, Thermo Scientific), as per the manufacturer’s instructions. Normalised samples were prepared in 5 × Laemmli buffer (50% glycerol, 5% SDS, 0.1% bromophenol blue, 250 mM Tris-HCL, pH 6.8) containing 5% β-mercaptoethanol and heated at 95 °C for 5 min, before SDS-polyacrylamide gel electrophoresis and electrotransfer to nitrocellulose (pore size: 0.2 µm; Amersham Protran Premium, Cytiva). Membranes were blocked with 5% non-fat milk in TBS containing 0.1% Tween^®^ 20 (TBS-T), incubated with a primary antibody overnight at 4 °C, and then incubated for 1-h with an appropriate secondary antibody. Membranes were washed in triplicate with TBS-T for 5 min between incubations. Immunoreactivity was visualised by film-based enhanced chemiluminescence (CL-XPosure™ Film™, Thermo Scientific) or by the Odyssey Imaging System (LI-COR Biosciences), depending on the nature of the conjugate.

### 2.5. Cellular Cytotoxicity Assay Panel

Cell viability was measured by the MTT proliferation assay. Briefly, 1 × 10^4^ cells per well were seeded in 96-well culture plates and incubated at 37 °C overnight. Subsequently, the cells were treated with various concentrations of pharmacological agents for 16-h as indicated; corresponding cells treated with dimethyl sulphoxide (DMSO) were used as a negative control; corresponding cells treated with Triton-X 100 were used as positive controls. Cells were carefully washed with 1× PBS and 20 µL of 5 mg/mL 3-(4,5-dimethylthiazol-2-yl)-2,5-diphenyl-2H-tetrazolium bromide (MTT) was added, and cells were incubated at 37 °C for a further 3-h. The media was removed, and the MTT-derived formazan product was solubilised in 100 µL of DMSO. The optical density of the solubilised formazan product was measured at 570 nm with a reference wavelength of 620 nm using an Infinite m200 spectrophotometer (Tecan, Männedorf, Switzerland).

### 2.6. ACE2 Enzyme-Linked Immunoassay (ELISA)

Total ACE2 levels in the conditioned media of HEK293T cells transiently expressing pcDNA3.1-hACE2 were quantified using an ACE2-specific ELISA kit (DuoSet ELISA, R&D Systems; catalogue number: DY933-0), as per the manufacturer’s instructions.

### 2.7. ACE2 C-Peptidase Activity Assay

ACE2 C-peptidase activity in the conditioned media of HEK293T cells transiently expressing pcDNA3.1-hACE2 was measured by cleavage of the quenched fluorescent substrate Mca-APK(Dnp)-OH (Sigma Aldrich; catalogue number: SML2948); fluorescence (excitation: 320 nm, emission: 405 nm) was measured in black-walled 96-well plates using an Infinite m200 spectrophotometer (Tecan, Männedorf, Switzerland).

### 2.8. Preparation of Artificial SARS-CoV-2 Conditioned Media (αSCM)

HEK293T cells transiently expressing pcDNA3.1-SARS2-Spike were cultured for 48-h post-transfection in complete DMEM, prior to the harvesting of conditioned media containing solubilised fragments of the SARS-CoV-2 spike protein (αSCM). To treat cells with αSCM, the medium was removed and replaced by an equal volume of αSCM for 2-h. Basal conditioned medium from parental untransfected HEK293T cells (pBCM) was collected as described above and used as a control medium.

To confirm the presence of the soluble SARS-CoV-2 S1 fragment in αSCM but not in pBCM, proteins were precipitated from the conditioned media by addition of ice-cold trichloroacetic acid to a final concentration of 20% (*w*/*v*). The precipitated proteins were separated from the supernatant by centrifugation (16,000× *g*, 30 min, 4 °C), and the supernatant discarded. The protein precipitates were washed in triplicate with acetone. Residual acetone in the samples was then removed by air-drying for 30 min at room temperature. Protein precipitates were solubilized in 1× Laemelli buffer before Western blot analysis.

### 2.9. In Vitro Cell–Cell Syncytiation Assays

HEK293T cells were transfected with pcDNA3.1-ACE2-C9 (HEK293T-ACE2), without or without co-transfection of pCSDest-TMPRSS2 or pRK5F-TACE, or pcDNA3.1-SARS2-Spike (HEK293T-SARS2) by calcium phosphate precipitation; cells were co-transfected with pcDNA3.1-eGFP or pcDNA3.1-mCherry, as indicated, for fluorescent reporter assays; cells were co-transfected with pCMV-α or pCMV-ω, as indicated, for β-galactosidase reporter assays. At 24-h post-transfection, target HEK293T-ACE2 cells and effector HEK293T-SARS2 cells were detached with 1 mM EDTA in 1× PBS (pH 7.5) and co-cultured in a ratio of 1:1 in 24-well plates. For pharmacological inhibition or potentiation of cell–cell fusion and syncytiation, pharmacological agents were added at the time of co-culture; unless directly indicated, concentrations are provided at the end of figure legends. For target cells and effector cells co-transfected with fluorescent reporters, syncytial co-cultures were counter-stained live with 1 µg/mL of Hoechst 33342 nuclear stain (Thermo Fisher Scientific) at 16-h post-co-culture and visualised on an EVOS FL Auto Imaging System (Thermo Fisher Scientific) using a 4× objective lens. For target cells and effector cells co-transfected with the β-galactosidase reporter, whole cell extracts of syncytial co-cultures were prepared at 16-h post-co-culture in 200 µL of lysis buffer (0.25 M Tris-HCl, 2.5 mM EDTA, 0.25% (*v*/*v*) Triton X-100) by continuous agitation at 450 RPM for 10 min at room temperature, prior to freezing at −80 °C. Plates were equilibrated at room temperature, and the whole cell extracts were clarified by centrifugation (16,000× *g*, 5 min, 4 °C); total protein concentration was determined by a BCA assay (Pierce™, Thermo Fisher), as per the manufacturer’s instructions. 10–20 μL of each whole cell extract was added to 200 μL of O-nitrophenyl-beta-D-galactopyranoside (ONPG; 4 mg/mL) and incubated at 37 °C. The reaction was stopped by the addition of 1 M sodium carbonate (Na_2_CO_3_) solution. The optical density of the yellow-coloured O-nitrophenol product was measured at 420 nm with a reference wavelength of 550 nm using an Infinite m200 spectrophotometer (Tecan, Männedorf, Switzerland). The activity of β-galactosidase was calculated as nanomoles of product formed per minute per milligram of lysate at 37 °C, according to Nielsen et al. [37].

### 2.10. Flow Cytometry

To confirm the surface expression of wild-type ACE2 and the ACE2 L584A mutant, transiently transfected HEK293T cells were harvested in 1 mM EDTA in 1× PBS (pH 7.5) and washed in FACS buffer (1% bovine serum albumin in 1× PBS (pH 7.5)). Cells were incubated on ice for 1-h with primary antibody, followed by three washes with FACS buffer. Cells were then incubated for 30 min with secondary antibody, followed by two washes with FACS buffer and resuspension in 1× PBS. Cells were analysed using an AccuriC6 flow cytometer (BD Biosciences). Primary and secondary antibodies were diluted in FACS buffer. Corrected mean fluorescence intensities were calculated by subtracting the mean fluorescence intensity with an isotype control antibody from the mean fluorescence intensity with the specific antibody.

### 2.11. Statistical Analysis

All measurements were derived from distinct samples and independent experiments. Data are typically presented as a pool of three experiments (mean ± standard deviation) or as a single experiment representative of two or more independent experiments (mean ± standard deviation). Data were compared using either the unpaired Student’s *t*-test, one-way ANOVA test with post hoc Dunnett test for multiple comparisons, or two-way ANOVA test with post hoc Tukey test for multiple comparisons; a *p*-value of less than 0.05 was considered significant (*, *p* < 0.05; **, *p* < 0.01; ***, *p* < 0.001; ****, *p* < 0.0001). Data were curated using Microsoft Excel (version 2108). All statistical analyses were performed on GraphPad Prism (version 9.00).

## 3. Results

### 3.1. SARS-CoV-2 Spike Protein-Induced Syncytiation Can Occur Independently of TMPRSS2

As the spike protein of SARS-CoV-2 induces cell–cell fusion and syncytiation when exogenously expressed in the host cell membrane during infection [21,26,27], we sought to establish a qualitative *in vitro* cell–cell fusion assay using transfected HEK293T cells to investigate TMPRSS2-independent factors involved in syncytiation (Figure 1A). Target cells transiently expressing the human ACE2 receptor (HEK293T-ACE2), with or without exogenous human TMPRSS2, were co-cultured with effector cells transiently expressing the SARS-CoV-2 spike protein (HEK293T-SARS2) to induce cell–cell fusion (Figure 1B); target and effector cells were also co-transfected with fluorescent proteins (HEK293T-ACE2-mCherry; HEK293T-SARS2-GFP) for qualitative live-cell monitoring of the extent of spike protein-induced syncytiation. Consistent with previous reports [14,17,18], spike protein-induced cell–cell fusion was ACE2-dependent and the expression of TMPRSS2 potentiated fusion (Figure 1C); however, considerable cell–cell fusion was also induced by the spike protein in the absence of TMPRSS2 (Figure 1C). This suggested the presence of a serine protease-independent pathway to syncytiation.

To exclude the involvement of alternative TMPRSS2-related serine proteases that may be expressed by HEK293T cells, we treated syncytial co-cultures with the serine protease inhibitor camostat mesylate [18,38]. Camostat mesylate prevented potentiation of spike protein-induced cell–cell fusion in the presence of TMPRSS2 and reduced fusion to a level similar to the DMSO-treated control; however, substantial syncytiation remained (Figure 1C). In contrast, camostat mesylate did not affect spike protein-induced cell–cell fusion in the absence of TMPRSS2, which further suggested a serine protease-independent pathway of syncytiation (Figure 1C). These findings show that the priming and activation of the SARS-CoV-2 spike protein is not exclusively dependent on the availability of host serine proteases, and that ancillary cellular proteases may facilitate cell–cell fusion and syncytiation in the absence of TMPRSS2.

### 3.2. Metalloproteases Promote SARS-CoV-2 Cell–Cell Fusion and Syncytiation

To investigate the proteases involved in TMPRSS2-independent syncytiation, we developed a quantitative *in vitro* cell–cell fusion assay based on a ‘split β-galactosidase’ reporter, as previously described by Theuerkauf et al. [36]. Target HEK293T-ACE2 cells transiently co-transfected with the α-fragment of β-galactosidase (HEK293T-ACE2-α) were co-cultured with effector HEK293T-SARS2 cells transiently co-transfected with the ω-fragment of β-galactosidase (HEK293T-SARS2-ω). Upon cell–cell fusion, α-complementation of the inactive ω-fragment resulted in the formation of active β-galactosidase and the fusion-associated β-galactosidase activity of whole cell extracts was quantitatively measured by O-nitrophenyl β-D-galactopyranoside (ONPG) cleavage (Figure 2A). Expression of the individual α- and ω-fragments in syncytial co-cultures did not result in the detection of β-galactosidase activity, and the assay confirmed ACE2-dependent and TMPRSS2-independent cell–cell fusion (Appendix A).

Syncytial co-cultures of target HEK293T-ACE2-α cells and effector HEK293T-SARS2-ω cells were next treated with a panel of protease inhibitors and the fusion-associated β-galactosidase activity measured to determine the mechanistic class of ancillary proteases involved in spike protein-induced cell–cell fusion. Consistent with a serine protease-independent pathway in HEK293T cells, spike protein-induced cell–cell fusion was largely unaffected by serine protease inhibitors, except for partial inhibition observed with the broad-spectrum inhibitor AEBSF (Figure 2B). Furthermore, no inhibition was observed with either leupeptin or pepstatin A, which inhibit primary serine/cysteine proteases and aspartic proteases, respectively (Figure 2B). Therefore, the involvement of such proteases in spike protein-induced cell–cell fusion was excluded from further study. Interestingly, fusion-associated β-galactosidase activity was reduced by the divalent metal ion chelators edetic acid (EDTA) and egtazic acid (EGTA), which suggested the involvement of a metalloprotease-dependent pathway in the formation of spike protein-induced syncytia (Figure 2B).

As the utility of EDTA and EGTA are limited by poor biocompatibility, particularly at higher concentrations which affect cellular adhesion and survival [34], we further investigated the inhibitory potential of hydroxamate-based metalloprotease inhibitors (batimastat, marimastat, and TAPI-1) against spike protein-induced cell–cell fusion. Hydroxamate-based metalloprotease inhibitors, which block several a disintegrin and metalloproteases (ADAMs) and matrix metalloproteases (MMPs), reduced fusion-associated β-galactosidase activity in a dose-dependent manner compared to the DMSO-treated control (Figure 2C). Dose-responsivity demonstrated inhibition of spike protein-induced cell–cell fusion by metalloprotease inhibitors in the low micromolar range, with over 75% loss of syncytiation compared to the DMSO-treated control at the concentration maxima. This reduction was attributed to inhibition of metalloprotease-dependent syncytiation as cellular viabilities were not affected by any of the inhibitors used (Appendix A) nor was the activity of β-galactosidase inhibited by the presence of inhibitors (Appendix A). Loss of spike protein-induced cell–cell fusion and syncytiation in the presence of metalloprotease inhibitors was also confirmed by fluorescent microscopy in syncytial co-cultures of target HEK293T-ACE2 cells and effector HEK293T-SARS2-GFP cells (Figure 2D). Next, the sensitivity of spike protein-induced cell–cell fusion to metalloprotease inhibition was examined in infection-competent Huh-7 cells, which endogenously express the ACE2 receptor but employ a TMPRSS2-independent mechanism of SARS-CoV-2 entry [39]. The α-fragment of β-galactosidase was transiently expressed in target Huh-7 cells and the resulting Huh-7-α transfectants were co-cultured with HEK293T-SARS2-ω cells in the presence or absence of the metalloprotease inhibitor TAPI-1, with or without the serine protease inhibitor camostat mesylate. Consistent with the utilisation of serine protease-independent pathways of SARS-CoV-2 entry in this cell line [39], TAPI-1 inhibited spike protein-induced cell–cell fusion in Huh-7 cells in a dose-dependent manner as shown by reduced fusion-associated β-galactosidase activity (Figure 2E); in contrast, camostat mesylate did not affect fusion-associated β-galactosidase activity in comparison to the DMSO-treated control (Figure 2E). Metalloprotease-dependent syncytiation in Huh-7 cells was also confirmed by fluorescent microscopy, wherein TAPI-1 but not camostat mesylate inhibited spike protein-induced cell–cell fusion when Huh-7 cells were co-cultured with HEK293T-SARS2-GFP cells (Figure 2F). Overall, these results indicate a distinct role for metalloproteases in SARS-CoV-2 spike protein-induced cell–cell fusion.

### 3.3. Metalloproteases Promote S2′-Activation of the SARS-CoV-2 Spike Protein

Given that cleavage of the SARS-CoV-2 spike protein is an essential requirement for fusion-competency [15,16], we next investigated which metalloproteases functionally ‘*prime*’ or ‘*activate*’ the viral fusogen. Among the metalloproteases blocked by hydroxamate-based inhibitors, the multifunctional sheddase ADAM17 (CD156b; TACE) is ubiquitously expressed in human tissues. ADAM17 mediates the ectodomain shedding of a wide range of different substrates, and its activity has been suggested to play a role in the inflammatory pathophysiology of SARS-CoV-2 [40]. Therefore, we considered ADAM17 as an ancillary protease for the SARS-CoV-2 spike protein, a hypothesis supported by the suboptimal conservation of amino acid residues of reported ADAM17 substrates [41] at the S2′ site (R815A), particularly the P5, P1, and P1′ positions.

To determine whether ADAM17 is involved in the cleavage of the SARS-CoV-2 spike protein, the metalloprotease activity of target HEK293T-ACE2-α cells co-cultured with effector HEK293T-SARS2-ω cells was augmented by either induction of endogenous ADAM17 activity with phorbol ester treatment [42] or transient overexpression of exogenous ADAM17. Increased ADAM17 activity correlated with increased fusion-associated β-galactosidase activity, which remained sensitive to metalloprotease inhibition (Figure 3A,B). Induction of endogenous ADAM17 activity with phorbol 12-myristate 13-acetate (PMA) was found to promote syncytiation to a greater extent than exogenous ADAM17 overexpression, which may be due to an increase in furin expression upon stimulation with phorbol esters [43]. Western blot analysis of the cleavage profile of the SARS-CoV-2 spike protein in HEK293T-SARS2 cells upon both induction of endogenous ADAM17 activity and overexpression of exogenous ADAM17 showed potentiated S2′-activation of the spike protein (Figure 3C,D). Furthermore, the increase in S2′-activation was dependent on metalloprotease activity as shown by reduction to the level of the DMSO-treated control when cells were pre-treated with TAPI-1 (Figure 3C,D). These results demonstrate that metalloproteases–in particular ADAM17–are novel SARS-CoV-2 S2′-activating proteases, which promote SARS-CoV-2 cell–cell fusion and syncytiation.

### 3.4. SARS-CoV-2 Variants of Concern Are Susceptible to Metalloprotease Inhibition

The persistent nature of the COVID-19 pandemic is largely dependent on the emergence of SARS-CoV-2 variants with increased virulence. To determine whether cell–cell fusion and syncytiation of current variants of concern remain susceptible to metalloprotease inhibition, we next investigated the cell–cell fusion of the Delta (B.1.617.2) and Omicron (B1.1.529; BA.1) variants in the presence of our hydroxamate-base inhibitors. Western blot analysis showed that the cleavage profile of the ancestral Wuhan-Hu-1 spike protein was similar to the cleavage profile of the Omicron variant (Figure 4A). However, the cleavage profile of the Delta variant showed a greater level of the primed S2 fragment (Figure 4A). Consistent with the enhanced priming associated with the Delta variant, we observed that expression of the spike protein of the Delta variant in HEK293T-ω cells was associated with greater fusion-associated β-galactosidase activity in comparison to the spike protein of the ancestral Wuhan-Hu-1 strain (Figure 4B); in contrast, despite similar a similar cleavage profile to the ancestral strain, the spike protein of the Omicron variant induced significantly less fusion (Figure 4B). As cell–cell fusion and syncytiation is thought to be important pathomechanism of SARS-CoV-2 [21,26,27], the reduced syncytiation may in part explain the lesser pathophysiology observed with the Omicron variant [44,45]. Nevertheless, SARS-CoV-2 cell–cell fusion mediated by the spike proteins of the Delta and Omicron variants, respectively, remained susceptible to metalloprotease inhibition (Figure 4C,D). As such, metalloprotease inhibitors may be efficacious not only against the ancestral SARS-CoV-2 strain but also against emerging variants of concern.

### 3.5. TMPRSS2-Expression Rescues SARS-CoV-2 Syncytiation in the Presence of Metalloprotease Inhibition

The identification of a metalloprotease-dependent pathway for SARS-CoV-2 cell–cell fusion and syncytiation prompted us to examine the effect of metalloprotease inhibition in the presence of an ancillary activating protease. It has been previously reported that ADAM17 and TMRPSS2 compete for substrates, including the SARS-CoV-2 receptor ACE2 [46], and both metalloproteases and serine proteases are co-expressed in several infection-competent cells. Target cells expressing the human ACE2 receptor (HEK293T-ACE2-α), with or without human TMPRSS2, were co-cultured with effector cells expressing the SARS-CoV-2 spike protein (HEK293T-SARS2-ω), and the fusion-associated β-galactosidase activity measured. As shown previously (Figure 1C), the presence of TMPRSS2 potentiated spike protein-induced cell–cell fusion and was suppressed to the level of the DMSO-treated control by the addition of camostat mesylate (Figure 5A); however, serine protease inhibition had no effect on cell–cell fusion in the absence of exogenous TMPRSS2 (Figure 5A). Moreover, the expression of TMPRSS2 in syncytial co-culture abrogated the inhibitory effect of hydroxamate-based metalloprotease inhibitors (Figure 5B), and significant inhibition of SARS-CoV-2 syncytiation was only observed when co-cultures were co-treated with inhibitors of both metalloproteases and serine proteases (Figure 5C). These findings were further investigated in Caco-2 and Calu-3 cells, human-derived lines highly susceptible to SARS-CoV-2 infection due to endogenous expression of both ACE2 and TMPRSS2 [47,48]. The α-fragment of β-galactosidase was transiently expressed in target Caco-2 and Calu-3 cells, respectively, and the resulting transfectants were co-cultured with HEK293T-SARS2-ω cells in the presence or absence of the metalloprotease inhibitor TAPI-1, with or without camostat mesylate (Figure 5D, 5F). The effect of metalloprotease and serine protease inhibitors on cell–cell fusion was also by investigated by fluorescent microscopy, whereby Caco-2 and Calu-3 cells were co-cultured with HEK293T-SARS2-GFP cells (Figure 5E,G). In contrast to our work in Huh-7 cells, metalloprotease inhibition was insufficient to reduce spike protein-induced cell–cell fusion in Caco-2 and Calu-3 cells when co-cultured with HEK293T-SARS2-ω cells, whereas treatment with camostat mesylate significantly reduced cell–cell fusion compared to the DMSO-treated control (Figure 5D,F). Inhibitors of both metalloproteases and serine proteases exhibited a synergistic effect on spike protein-induced cell–cell fusion (Figure 5F) and eliminated syncytiation, as shown by the complete loss of syncytiation under fluorescent microscopy when TAPI-1 and camostat mesylate were co-administered (Figure 5E,G). These results suggest that inhibition of a single mechanistic class of activating proteases may not be sufficient to prevent SARS-CoV-2 cell–cell fusion and syncytiation *in vitro* nor *in vivo* as metalloproteases and serine proteases are likely to co-operatively compensate for single-mechanistic inhibition.

### 3.6. SARS-CoV-2 Spike Protein Induces ACE2 Ectodomain Shedding

Under normal physiological conditions, soluble ACE2 is not detectable in the peripheral circulation [49,50]; however, it is significantly elevated in patients with COVID-19 [51,52,53,54]. ACE2 is released from the cell surface in a process of ectodomain shedding mediated by proteases which also cleave the SARS-CoV-2 spike protein, namely ADAM17 and TMPRSS2 [46]. It has been reported that the interaction of the SARS-CoV-2 spike protein with the ACE2 receptor triggers a conformational change that facilitates proteolytic cleavage of the spike protein [16]. Western blot analysis of the spike protein in whole cell extracts of target HEK293T transiently expressing human ACE2 C-terminally tagged with EGFP (HEK293T-ACE2e) co-cultured with effector HEK293T-SARS2 showed the expression of the full-length spike protein and the primed S2 fragment, and confirmed the ACE2-dependent potentiation of S2′-activation (Figure 6A). Therefore, we considered the possibility that the interaction of the SARS-CoV-2 spike protein with ACE2 may trigger a reciprocal cleavage event, whereby the priming and activation of the spike protein is coupled to the shedding of the ACE2 receptor and the increased soluble ACE2 seen in patients with COVID-19. Western blot analysis of the ACE2 receptor in whole cell extracts of target HEK293T-ACE2e cells co-cultured with effector HEK293T-SARS2 cells revealed the expression of the full-length ACE2 protein and the membrane-bound C-terminal fragment (CTF) generated by ectodomain shedding [55], and that the generation of the ACE2 CTF was also potentiated by the presence of the SARS-CoV-2 spike protein (Figure 6B). The increased production of the ACE2 CTF was found to be metalloprotease-dependent susceptible to TAPI-1 (Figure 6B). The shedding of the ACE2 ectodomain into the conditioned media was confirmed by an ACE2-ELISA (Figure 6C) and the solubilised form of ACE2 was shown to be catalytically active by cleavage of an ACE2-specific fluorogenic substrate (Figure 6D). To further investigate the effect of the SARS-CoV-2 spike protein on the processing of ACE2, we generated artificial SARS-CoV-2 conditioned media (αSCM), defined as spent medium collected from HEK293T-SARS2 cells containing solubilised components of the SARS-CoV-2 spike protein (Figure 6E); basal conditioned medium (pBCM), which consisted of spent medium collected from parental HEK293T cells, served as the negative control. Western blot analysis of trichloroacetic acid-precipitated proteins from the αSCM revealed an immunoreactive band consistent with the molecular weight of the SARS-CoV-2 S1 subunit released from the SARS-CoV-2 spike protein following cleavage at the S1/S2 junction (Figure 6F). HEK293T-ACE2 cells exposed to αSCM exhibited increased shedding of ACE2 from the cell surface, as shown by increased formation of the membrane-bound ACE2 C-terminal fragment (Figure 6F). In line with the spike protein-induced cleavage profile of ACE2, increased ACE2 concentrations and ACE2 activity was observed in the conditioned media of HEK293T-ACE2 cells exposed to αSCM (Figure 6G,H). Shedding of ACE2 from host cells in response to αSCM was also found to be metalloprotease-dependent, with a reduction in the level of the ACE2-CTF in whole cell extracts in the presence of the hydroxamate-based metalloprotease inhibitor TAPI-1 (Figure 6I). These results suggest that the SARS-CoV-2 spike protein can induce metalloprotease-dependent ACE2 ectodomain shedding, in a reciprocal manner with ACE2-associated SARS-CoV-2 priming.

### 3.7. Genetic Attenuation of ACE2 Ectodomain Shedding Reduces SARS-CoV-2 Syncytiation

As we have shown that the SARS-CoV-2 spike protein induces ACE2 ectodomain shedding, and previous studies have implicated ADAM17-ACE2 interactions as modulators of SARS-CoV-1 [56,57,58], we next investigated whether the shedding of ACE2 contributed to SARS-CoV-2 spike protein-induced cell–cell fusion *in vitro*. For this investigation, we used a single point mutation in the ACE2 receptor, L584A, positioned within a proposed protease recognition sequence and reported to markedly attenuate ectodomain shedding [59]. Western blot analysis showed that the ACE2 L584A mutant expressed similarly to wild-type ACE2 (Figure 7A), and flow cytometric analysis confirmed that the mutation did not affect trafficking to the cell surface (Figure 7B). In contrast to wild-type ACE2, however, an ACE2 L584A CTF was not detectable by Western blot analysis under conditions of constitutive or enhanced metalloprotease activity (Figure 7A). The ACE2 L584A mutant was confirmed to have greater resistance to phorbol ester- and ADAM17-induced ectodomain shedding, as shown by ELISA and activity assays (Figure 7C,D), which confirmed attenuated cleavability [59].

Inclusion of the ACE2 L584A mutant into the *in vitro* cell–cell fusion assay revealed a significant reduction in spike protein-induced syncytiation when compared to wild-type ACE2, as measured by reduced fusion-associated β-galactosidase activity (Figure 7E). The remaining spike protein-induced cell–cell fusion of the ACE2 L584A mutant was susceptible to hydroxamate-based metalloprotease inhibition (Figure 7F), which is likely due to the aforementioned inhibition of metalloprotease-dependent S2′-activation of the SARS-CoV-2 spike protein. This result suggests that ACE2 ectodomain shedding potentially contributes to, but is not essential for, cell–cell fusion. Given that the presence of TMPRSS2 abrogated sensitivity to metalloprotease inhibition, we examined the effect of TMPRSS2 co-expression on the ACE2 L584A mutant. Western blot analysis showed that the ACE2 L584A mutant was not resistant to cleavage by TMPRSS2 (Figure 7G), as ADAM17 and TMPRSS2 cleave ACE2 differentially at distinct locations within the peri-transmembrane region [46]. TMPRSS2-mediated cleavage of the ACE2 L584A mutant was confirmed by increased soluble ACE2 concentrations and ACE2 activity in the conditioned media of HEK293T cells transiently overexpressing the ACE2 L584A mutant (Figure 7H,I). The presence of the serine protease TMPRSS2 rescued the syncytial capacity of HEK293T-ACE2-L584A cells (Figure 7J), suggesting an importance for ACE2 proteolysis. Overall, these findings demonstrate that SARS-CoV-2-induced ectodomain shedding may promote spike protein-induced cell–cell fusion and syncytiation, identifying a novel mechanism contributing to SARS-CoV-2 pathophysiology.

## 4. Discussion

SARS-CoV-2 spike protein-induced cell–cell fusion and syncytiation during the infectious life cycle represents a considerable pathomechanism associated with the severity of COVID-19. Syncytiation is dependent on the inherent fusogenic capacity of the SARS-CoV-2 spike protein, which requires proteolytic processing to achieve fusion-competency. In this study, we have identified a metalloprotease-dependent pathway contributing to SARS-CoV-2 cell–cell fusion and syncytiation in the absence of serine proteases. Furthermore, we mechanistically show that the metalloprotease-dependent pathway involves S2′-activation of the SARS-CoV-2 spike protein, and that reciprocal metalloprotease-dependent cleavage of the ACE2 receptor, in response to the engagement of the viral spike protein, may potentiate SARS-CoV-2 syncytiation.

It has been commonly reported that proteolytic processing of the SARS-CoV-2 spike protein is dependent on the activity of TMPRSS2 and TMPRSS2-related serine proteases within the respiratory system [9,17,18,20]. SARS-CoV-2 infection in humans typically begins in cells of the upper respiratory tract, which express the ACE2 receptor and the S2′-activating protease TMPRSS2 [60], prior to dissemination to the lower respiratory tract. However, the expression of TMPRSS2 is significantly less in sites of dissemination [61,62], and raises the possibility of alternative SARS-CoV-2 activating proteases. Recent studies have also shown the presence of alternative pathways for SARS-CoV-2 syncytiation that are independent of serine proteases [33,34].

We report herein that the metalloprotease ADAM17 promotes SARS-CoV-2 cell–cell fusion by S2′-activation of the SARS-CoV-2 spike protein, which exposes the highly conserved fusion peptide. ADAM17 has been previously implicated in the pathophysiology of SARS-CoV-1 [56,57,58], which is the most closely related human coronavirus to SARS-CoV-2 [63,64]. Antagonists of ADAM17 were shown to inhibit SARS-CoV-1 infectivity *in vitro* and *in vivo* [57,58]. Our findings complement and extend a recent study by Jocher et al. [65] that reported the promotion of SARS-CoV-2 cell entry by ADAM17 and ADAM10 in the absence of serine proteases. In support of the cleavage of recombinant SARS-CoV-2 protein at the S2′ site outlined by Jocher et al. [65], we confirm herein ADAM17-dependent S2′-activation in cultured cells. However, although both studies implicate ADAM17 in the pathophysiology of SARS-CoV-2, the mechanism is reportedly different: our findings show that ADAM17-dependent S2′-activiation promotes SARS-CoV-2 syncytiation under conditions of enhanced ADAM17 activity, whereas Jocher et al. [65] reported that ADAM17-knockout reduced infectivity but did not affect SARS-CoV-2 syncytiation. Differences in experimental design, such as knockout versus overexpression of ADAM17, may partly explain the discrepancies that exist between these studies. A further study by Yamamoto et al. [66] recently reported that ADAM17-knockout failed to reduce SARS-CoV-2 pseudovirus entry by any means but implicated ADAM10 in viral infectivity. As such, although metalloproteases are emerging as critical contributors to SARS-CoV-2 pathophysiology, the precise nature and involvement of metalloproteases in SARS-CoV-2 infection is likely to require further research.

In the context of co-expression, metalloproteases and serine proteases act cooperatively in the S2′-activation of the SARS-CoV-2 spike protein. In the presence of serine proteases, such as TMPRSS2, hydroxamate-based metalloprotease inhibitors did not affect SARS-CoV-2 cell–cell fusion; while serine protease inhibition attenuated SARS-CoV-2 syncytiation, complete inhibition could only be achieved by cotreating with inhibitors of the metalloprotease- and serine protease-dependent pathways of SARS-CoV-2 cell–cell fusion. Previous reports have indicated a potential clinical benefit of camostat mesylate administration to patients with COVID-19 [18,38,67]; however, evidence from randomised trials suggest that inhibition of serine proteases did not affect the severity and duration of moderate to severe COVID-19 [68]. This may be partially explained by the compensatory role of metalloproteases and serine proteases demonstrated in this study.

To our knowledge, this is the first study to implicate SARS-CoV-2 spike protein-induced shedding of the ACE2 receptor as a contributory factor to SARS-CoV-2 cell–cell fusion and syncytiation. Under normal physiological conditions, soluble ACE2 is difficult to detect in patient plasma [49,50]; however, in patients with severe COVID-19, increased concentrations of soluble ACE2 are observed which correlate with pathoprogression [51,52,53,54]. Previous studies have shown that SARS-CoV-1 induces ACE2 ectodomain shedding [56,58], and we demonstrate that the SARS-CoV-2 spike protein induces the shedding of a catalytically active ACE2 ectodomain from the surface of host cells. Moreover, shedding of the ACE2 receptor by TMPRSS2 has been implicated in the infectivity of SARS-CoV-1 [46], and recently a TMPRSS2-resistant ACE2 mutant was shown to attenuate SARS-CoV-2 infectivity *in vitro* [69]. Using a previously reported ACE2 mutant resistant to ADAM17-dependent ectodomain shedding [59], we demonstrate that SARS-CoV-2 spike protein-induced cell–cell fusion and syncytiation is attenuated in the absence of ADAM17-mediated ACE2 proteolysis. The residual syncytial capacity of cells expressing the ACE2 L584A mutant was shown to be metalloprotease-dependent, which supports the emerging role of compensatory metalloproteases in SARS-CoV-2 S2′-activation [65]. We hypothesise that the shedding of the ACE2 ectodomain may mechanically facilitate the dissociation of the S1 subunit of the SARS-CoV-2 spike protein, in support of a recent report that the spike protein may be mechanically activated by force-induced detachment of the S1 subunit [70].

Previous reports have suggested that concentrations of soluble ACE2 in excess of the physiological range may inhibit SARS-CoV-2 infectivity [65,71,72], such that ACE2 shedding from the cell surface may be considered a pathoprotective and desired process. In contrast, it has been proposed that soluble ACE2 concentrations closer to the physiological context may in fact enhance cellular entry of SARS-CoV-2 [73], as soluble ACE2 can bind the SARS-CoV-2 spike protein and facilitate endocytic entry via components of the renin-angiotensin system [74]. Therefore, identification of metalloprotease-dependent spike protein-induced ACE2 shedding as a contributor to SARS-CoV-2 cell–cell fusion supports the inhibition of metalloproteases as a treatment for patients with COVID-19.

Overall, targeting host metalloproteases in patients with COVID-19 may represent a novel therapeutic modality. Most patients with COVID-19 do not develop a severe pathological phenotype; however, extreme forms of COVID-19 associated with significant morbidity and mortality are characterised by a systemic hyperinflammatory response, venous thromboembolism, and multi-organ failure [75,76]. Metalloproteases have a longstanding link to the progression of respiratory damage [77], and MMP-2 [78], MMP-3 [79], MMP-7 [80], and MMP-9 [78,79,81] have all been shown to correlate with COVID-19 severity. ADAM17 is an emerging therapeutic target for several pulmonary pathologies [82,83,84], and its inhibition has been shown to prevent the pathophysiological neutrophilia associated with COVID-19 [85,86,87].

In conclusion, the findings outlined in this study support an emerging appreciation that SARS-CoV-2 S2′-activation is not entirely serine protease-dependent [33,34,65,66], and that metalloproteases act co-operatively and interchangeably to expand the tropism of SARS-CoV-2. We contribute to the mechanistic understanding of SARS-CoV-2 cell–cell fusion and syncytiation and suggest novel therapeutic stratagems involving multi-class protease inhibitors may be beneficial for the treatment of patients with COVID-19.

## Figures and Tables

**Figure 1 viruses-14-02094-f001:**
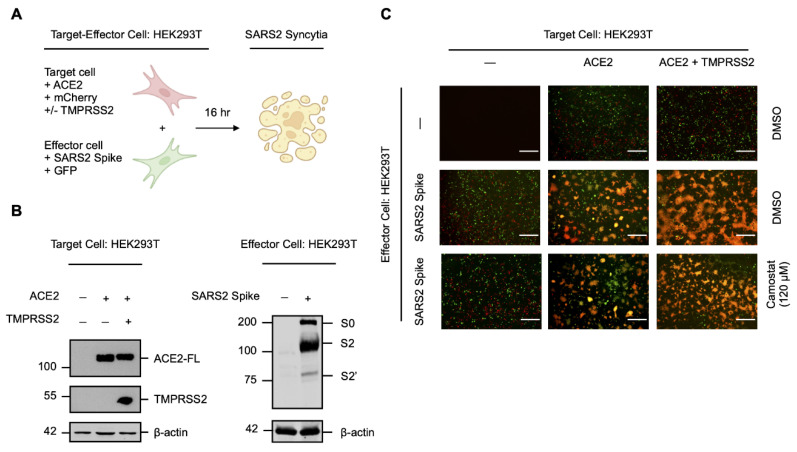
Serine protease-independent SARS-CoV-2 cell–cell fusion and syncytiation. (**A**) Experimental schematic of a qualitative *in vitro* cell–cell fusion assay with fluorescent reporters. Target HEK293T cells were transiently co-transfected with human ACE2 and mCherry (HEK293T-ACE2-mCherry), with or without human TMPRSS2, and were co-cultured with effector HEK293T cells transiently co-transfected with the SARS-CoV-2 (SARS2) spike protein and GFP (HEK293T-SARS2-GFP). (**B**) Western blot analysis of ACE2 and TMPRSS2 in HEK293T-ACE2-mCherry cells and the SARS2 spike protein in HEK293T-SARS2-GFP cells. ACE2 and the SARS2 spike protein were immunodetected with an anti-1D4 rhodopsin antibody, directed against the C-terminal C9-tag; TMPRSS2, with an anti-TMPRSS2-specific antibody. Western blotting data are from one experiment representative of at least three independent experiments. Immunodetection of β-actin served as a loading control. (**C**) Representative fluorescent micrographs of HEK293T-ACE2-mCherry and HEK293T-SARS2-GFP co-cultures at 16-h post-co-culture, in the presence or absence of the serine protease inhibitor camostat mesylate (120 µM; 16-h). ACE2-FL: full-length ACE2; S0: full-length SARS-CoV-2 spike protein; S2: S2 fragment of the SARS-CoV-2 spike protein; S2′: S2′ fragment of the SARS-CoV-2 spike protein. Scale bar: 500 μm.

**Figure 2 viruses-14-02094-f002:**
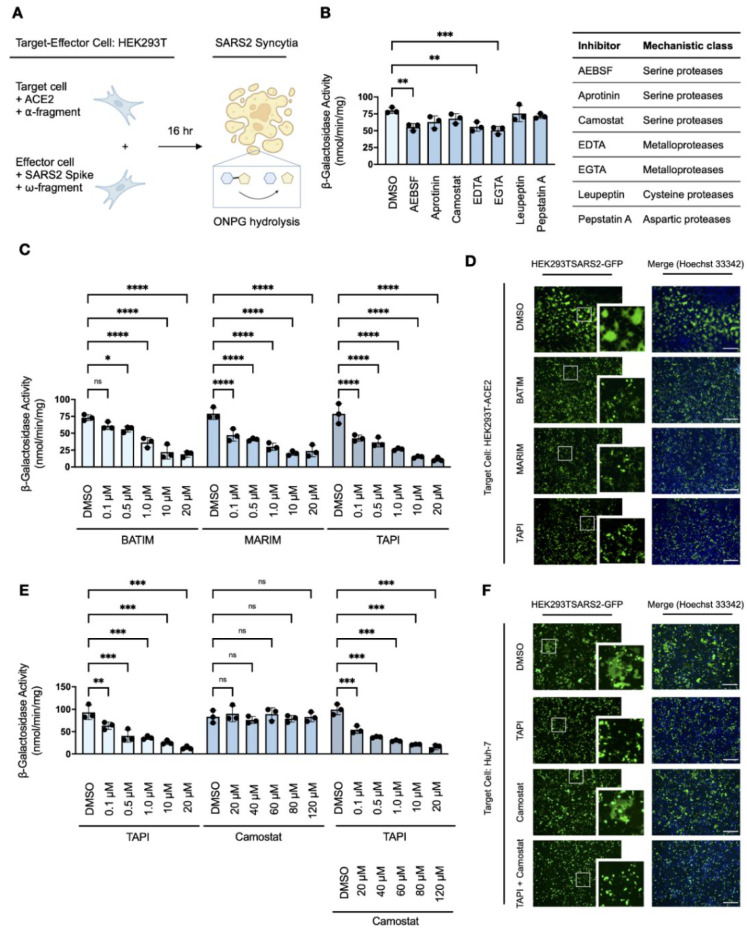
Metalloprotease-dependent syncytiation of the SARS-CoV-2 spike protein (**A**) Experimental schematic of the quantitative *in vitro* cell–cell fusion assay with split β-galactosidase reporter. (**B**) Effects of protease inhibitors (AEBSF, 1 mM; aprotinin, 10 μM; camostat mesylate, 120 μM; EDTA, 2.5 mM; EGTA, 2.5 mM; leupetin, 10μM; pepstatin A, 10 μM; 16-h) on fusion-associated β-galactosidase-activity of HEK293T-ACE2-⍺ and HEK293T-SARS2-ω co-cultures (one-way ANOVA; Dunnett post hoc). (**C**) Effects of hydroxamate-based metalloprotease inhibitors on fusion-associated β-galactosidase-activity of HEK293T-ACE2-⍺ and HEK293T-SARS2-ω co-cultures (two-way ANOVA; Tukey post hoc). (**D**) Representative fluorescent micrographs of HEK293T-ACE2 and HEK293T-SARS2-GFP co-cultures post-treatment with hydroxamate-based metalloprotease inhibitors. (**E**). Effects of hydroxamate-based metalloprotease inhibitors on fusion-associated β-galactosidase-activity of Huh-7-⍺ and HEK293T-SARS2-ω co-cultures (two-way ANOVA; Tukey post hoc). (**F**) Representative fluorescent micrographs of Huh-7 and HEK293T-SARS2-GFP co-cultures post-treatment with TAPI-1 and/or camostat mesylate. Syncytiation was visualised at 16-h post-co-culture on an EVOS FL Auto Imaging System (Thermo Fisher Scientific). Camostat: Camostat mesylate (120 μM; 16-h); BATIM: Batimastat (20 μM; 16-h); MARIM: Marimastat (20 μM; 16-h); TAPI: TAPI-1 (20 μM; 16-h); SARS2: SARS-CoV-2 spike protein. Probability value: *, *p* < 0.05; **, *p* < 0.01; ***, *p* < 0.001; ****, *p* < 0.0001. Scale bar: 500 μm. Fluorescent micrographs are also available in Appendix A.

**Figure 3 viruses-14-02094-f003:**
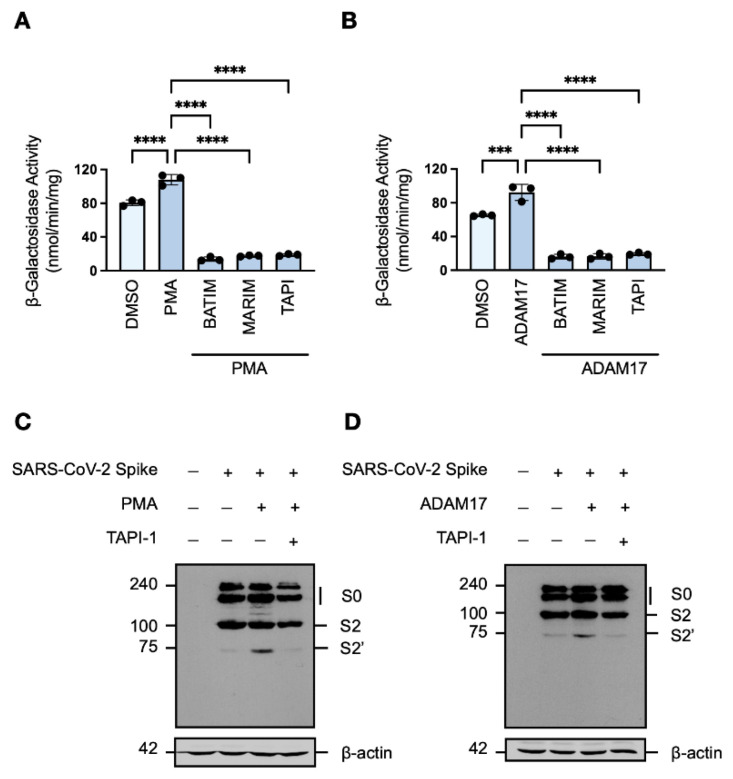
ADAM17 promotes S2′-activation of the SARS-CoV-2 spike protein. (**A**,**B**) Effects of phorbol ester-induced endogenous ADAM17 activity or exogenous ADAM17 overexpression on fusion-associated β-galactosidase-activity of HEK293T-ACE2-⍺ cells co-cultured with HEK293T-SARS2-ω cells, with or without hydroxamate-based metalloprotease (two-way ANOVA; Tukey post hoc). (**C**,**D**) Western blot analysis of the SARS-CoV-2 spike protein (SARS2) under conditions of phorbol ester-induced ADAM17 activity or exogenous ADAM17 overexpression, with or without TAPI-1. The SARS-CoV-2 spike protein (SARS2) was immunodetected with an anti-1D4 rhodopsin antibody, directed against the C-terminal C9-tag. Western blotting data are from one experiment representative of at least three independent experiments. Immunodetection of β-actin served as a loading control. PMA: Phorbol 12-myristate 13-acetate (200 ng/mL; 2-h) BATIM: Batimastat (20 μM; 16-h); MARIM: Marimastat (20 μM; 16-h); TAPI: TAPI-1 (20 μM; 16-h); S0: full-length SARS-CoV-2 spike protein; S2: S2 fragment of the SARS-CoV-2 spike protein; S2′: S2′ fragment of the SARS-CoV-2 spike protein. Probability value: ***, *p* < 0.001; ****, *p* < 0.0001.

**Figure 4 viruses-14-02094-f004:**
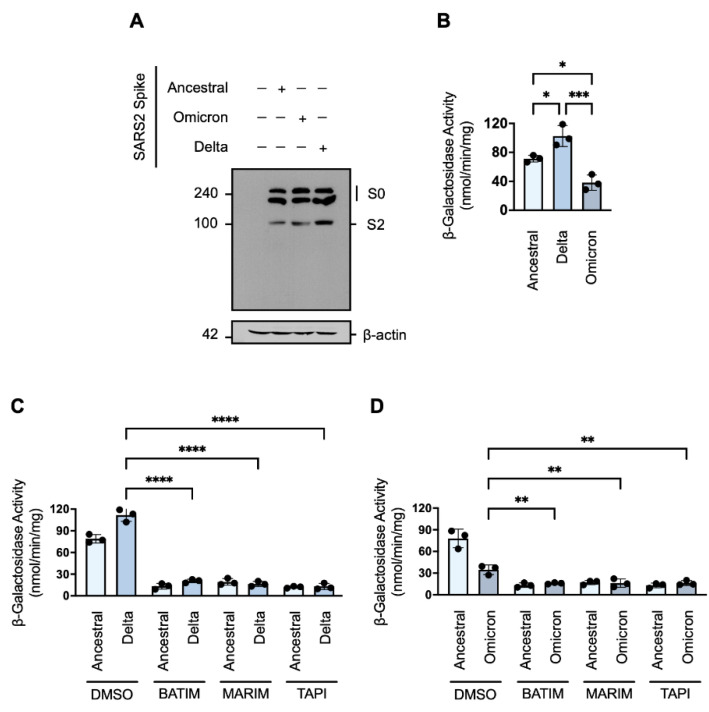
SARS-CoV-2 variants of concern are susceptible to metalloprotease inhibition. (**A**) Western blot analysis of the SARS-CoV-2 spike protein (SARS2) of the ancestral Wuhan-Hu-1 strain or the Omicron (B.1.1.5129) or Delta (B.1.617.2) variants in transiently transfected HEK293T cells. The SARS-CoV-2 spike protein of the ancestral Wuhan-Hu-1 strain was immunodetected with an anti-1D4 rhodopsin antibody, directed against the C-terminal C9-tag, and the spike proteins of the SARS-CoV-2 Omicron (B.1.1.5129) or Delta (B.1.617.2) variants were immunodetected with an anti-FLAG antibody, directed against the C-terminal FLAG tag. Western blotting data are from one experiment representative of at least three independent experiments. Immunodetection of β-actin served as a loading control. (**B**) Comparison of the fusion-associated β-galactosidase-activity of HEK293T-ACE2-⍺ and HEK293T-SARS2-ω co-cultures, wherein HEK293T-SARS2-ω express either the ancestral Wuhan-Hu-1 strain, the Omicron (B.1.5129), Delta (B.1.617.2) SARS-CoV-2 spike protein (one-way ANOVA; Dunnett post hoc). (**C**,**D**) Effects of hydroxamate-based metalloprotease inhibitors on fusion-associated β-galactosidase-activity of HEK293T-ACE2-⍺ and HEK293T-SARS2-ω co-cultures, wherein HEK293T-SARS2-ω express either the Delta (B.1.617.2) or Omicron (B.1.5129) (two-way ANOVA; Tukey post hoc). BATIM: Batimastat (20 μM; 16-h); MARIM: Marimastat (20 μM; 16-h); TAPI: TAPI-1 (20 μM; 16-h); SARS2: SARS-CoV-2 spike protein; S0: full-length SARS-CoV-2 spike protein; S2: S2 fragment of the SARS-CoV-2 spike protein. Probability value: *, *p* < 0.05; **, *p* < 0.01; ***, *p* < 0.001; ****, *p* < 0.0001.

**Figure 5 viruses-14-02094-f005:**
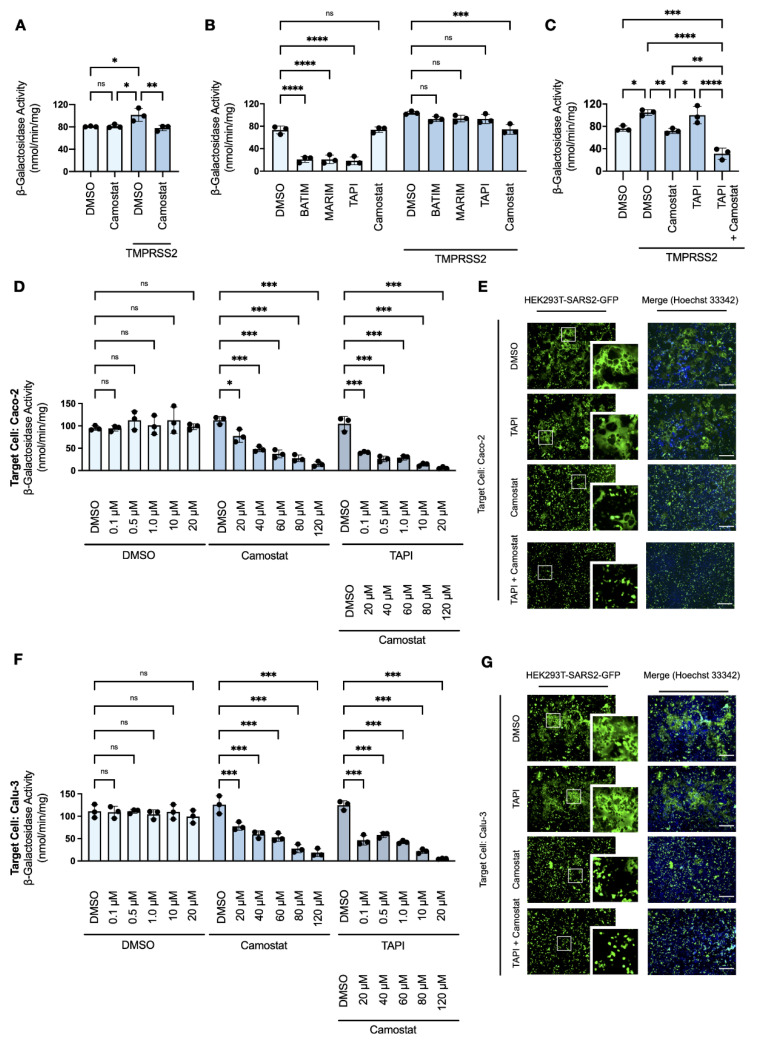
TMPRSS2-dependent SARS-CoV-2 cell–cell fusion and syncytiation is resistant to metalloprotease inhibition. (**A**) Effect of TMPRSS2 overexpression on fusion-associated β-galactosidase activity of HEK293T-ACE2-⍺ and HEK293T-SARS2-ω co-cultures in the presence or absence of the serine protease inhibitor camostat mesylate, or (**B**) hydroxamate-based metalloprotease inhibitors (two-way ANOVA; Tukey post hoc). (**C**) Effect of TAPI-1 and/or camostat mesylate on fusion-associated β-galactosidase activity of HEK293T-ACE2-⍺, with or without TMPRSS2, and HEK293T-SARS2-ω co-cultures (two-way ANOVA; Tukey post hoc). (**D**) Effects of TAPI-1 and/or camostat mesylate on fusion-associated β-galactosidase activity of Caco-2-⍺ and HEK293T-SARS2-ω co-cultures (two-way ANOVA; Tukey post hoc). (**E**) Representative fluorescent micrographs of Caco-2 and HEK293T-SARS2-GFP co-cultures post-treatment with TAPI-1 and/or camostat mesylate. (**F**) Effects of TAPI-1 and/or camostat mesylate on fusion-associated β-galactosidase activity of Calu-3-⍺ and HEK293T-SARS2-ω co-cultures (two-way ANOVA; Tukey post hoc). (**G**) Representative fluorescent micrographs Calu-3 and HEK293T-SARS2-GFP co-cultures post-treatment with TAPI-1 and/or camostat mesylate. Syncytiation was visualised at 16-h post-co-culture on an EVOS FL Auto Imaging System (Thermo Fisher Scientific). Camostat mesylate (120 μM; 16-h); BATIM: Batimastat (20 μM; 16-h); MARIM: Marimastat (20 μM; 16-h); TAPI: TAPI-1 (20 μM; 16-h); SARS2: SARS-CoV-2 spike protein. Probability value: *, *p* < 0.05; **, *p* < 0.01; ***, *p* < 0.001; ****, *p* < 0.0001. Scale bar: 500 μm. Fluorescent micrographs are also available in Appendix A (Appendix A).

**Figure 6 viruses-14-02094-f006:**
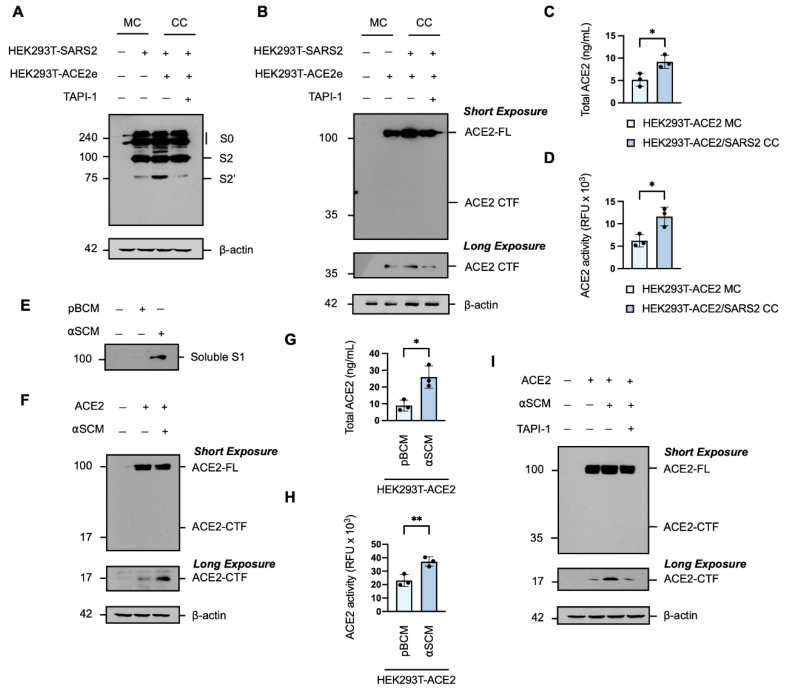
The SARS-CoV-2 spike protein induces ACE2 ectodomain shedding. (**A**). Western blot analysis of the SARS-CoV-2 spike protein and (**B**) ACE2 in monocultures (MC) or co-cultures (CC) of HEK293T cells overexpressing ACE2, with a C-terminal eGFP-tag (HEK293T-ACE2e), and SARS-CoV-2 spike protein (HEK293T-SARS2), in the presence of absence of TAPI-1. (**C**) Total soluble ACE2 and (**D**) relative ACE2 C-peptidase activity in the conditioned media of cultures in (**B**) (unpaired Student’s *t*-test). (**E**) Western blot analysis of the soluble S1 fragment precipitated from artificial SARS-CoV-2 conditioned media (αSCM) of HEK293T-SARS2 cells; parental basal conditioned media (pBCM) from untransfected cells served as a negative control. (**F**) Western blot analysis of ACE2 in HEK293T-ACE2e incubated with αSCM (2-h). (**G**) Total ACE2 determined by ELISA and (**H**) relative ACE2 C-peptidase activity in the conditioned media of HEK293T-ACE2-GFP incubated with αSCM (2-h) (unpaired Student’s *t*-test). (**I**) Western blot analysis of ACE2 in HEK293T-ACE2-GFP incubated with αSCM, in the presence of absence of TAPI-1. The SARS-CoV-2 spike protein (SARS2) was immunodetected with an anti-1D4 rhodopsin antibody, directed against the C-terminal C9-tag; ACE2, with an anti-eGFP antibody, directed against the C-terminal eGFP-tag. Western blotting data are from one experiment representative of at least three independent experiments. Immunodetection of β-actin served as a loading control. TAPI: TAPI-1 (20 μM; 16-h) SARS2: SARS-CoV-2 spike protein; S0: full-length SARS-CoV-2 spike protein; S2: S2 fragment of the SARS-CoV-2 spike protein. ACE2-FL: full-length ACE2; ACE2-CTF: ACE2 C-terminal fragment. Probability value: *, *p* < 0.05; **, *p* < 0.01.

**Figure 7 viruses-14-02094-f007:**
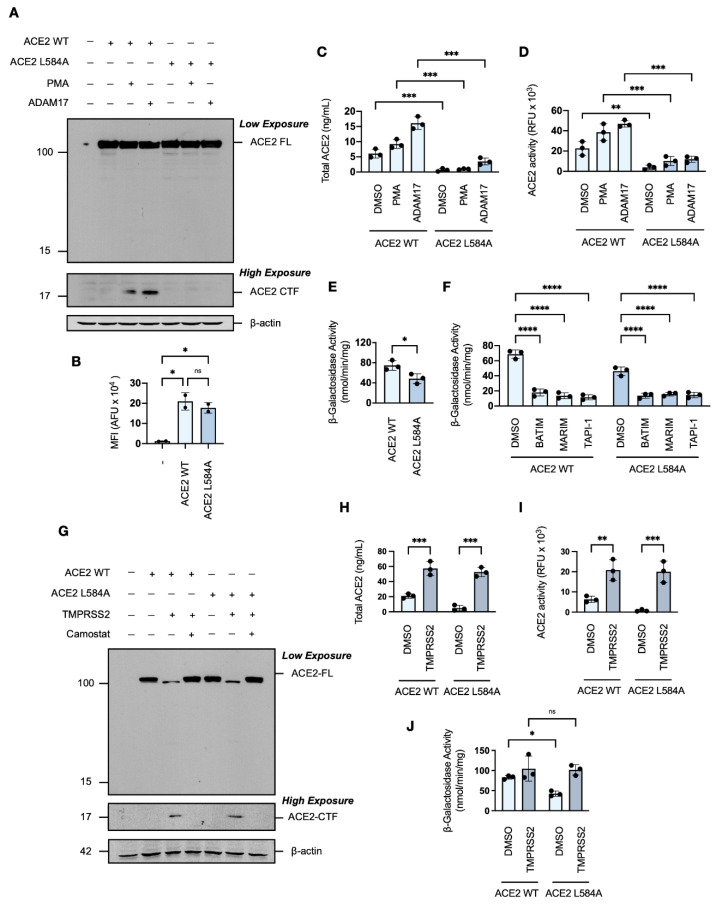
Metalloprotease-dependent ectodomain shedding of ACE2 contributes to SARS-CoV-2 cell–cell fusion and syncytiation. (**A**) Western blot analysis of wild type ACE2 and the ACE2 L584A mutant transiently overexpressed in HEK293T cells, in the presence or absence of phorbol ester stimulation or transient exogenous ADAM17 co-transfection. (**B**) Median fluorescence intensity (MFI) in arbitrary fluorescent units (AFU) of ACE2 on the surface of HEK293T cells transiently transfected with ACE2 or the ACE2 L584A mutant (unpaired Student’s *t*-test). (**C**) Total ACE2 determined by ELISA and (**D**) relative ACE2 C-peptidase activity in the conditioned media of HEK293T cells transiently transfected with human with ACE2 or the ACE2 L584A mutant, as described in (**A**) (two-way ANOVA; Tukey post hoc). (**E**) Effects of the ACE2 L584A mutation on fusion-associated β-galactosidase activity in co-cultures of target HEK293T-ACE2-⍺ cells or HEK293T-ACE2-L584A-⍺ and effector HEK293T-SARS2-ω cells (unpaired Student’s *t*-test). (**F**) Effect of the ACE2 L584A mutation on fusion-associated β-galactosidase activity in co-cultures of target HEK293T-ACE2-⍺ cells or HEK293T-ACE2-L584A-⍺ and effector HEK293T-SARS2-ω cells with or without hydroxamate-based metalloprotease inhibitors (two-way ANOVA; Tukey post hoc). (**G**) Western blot analysis of wild type ACE2 and the ACE2 L584A mutant, in the presence or absence of serine protease inhibition and/or transient TMPRSS2 co-transfection. (**H**) Total ACE2 determined by ELISA and (**I**) relative ACE2 C-peptidase activity in the conditioned media of HEK293T cells transiently transfected with human with ACE2 or the ACE2 L584A mutant, as described in (**G**) (two-way ANOVA; Tukey post hoc). (**J**) Effect of the ACE2 L584A mutation on fusion-associated β-galactosidase activity in co-cultures of target HEK293T-ACE2-⍺ cells or HEK293T-ACE2-L584A-⍺ and effector HEK293T-SARS2-ω cells, with or without the transient co-transfection of TMPRSS2 (two-way ANOVA; Tukey post hoc). ACE2 and the ACE2 L584A mutant were immunodetected with an anti-1D4 rhodopsin antibody, directed against the C-terminal C9-tag. Western blotting data are from one experiment representative of at least three independent experiments. Immunodetection of β-actin served as a loading control. PMA: Phorbol 12-myristate 13-acetate (200 ng/mL; 2-h); Camostat mesylate (120 μM; 16-h); BATIM: Batimastat (20 μM; 16-h); MARIM: Marimastat (20 μM; 16-h); TAPI: TAPI-1 (20 μM; 16-h); ACE2-FL: full-length ACE2; ACE2-CTF: ACE2 C-terminal fragment. Probability value: *, *p* < 0.05; **, *p* < 0.01; ***, *p* < 0.001; ****, *p* < 0.0001.

## Data Availability

The data presented in this study are available on reasonable request from the corresponding author.

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
