# Peer review of "Metalloprotease-Dependent S2′-Activation Promotes Cell–Cell Fusion and Syncytiation of SARS-CoV-2"

_viruses, 2022, doi:10.3390/v14102094_

Round 1

Reviewer 1 Report

Harte et al. investigated the role of metalloproteases on cell-cell fusion induced by the SARS-CoV-2 Spike protein. They used co-cultures of S+ and ACE2+ transiently transfected cells treated with various protease inhibitors and quantified syncytia using a β-galactosidase complementation assay. They first show that broadly inhibiting metalloproteases in TMPRSS2 negative cells inhibits Spike mediated cell-cell fusion. This inhibition is lost when TMPRSS2 is present. These results suggest that metalloproteases are important priming factors of S in TMPRSS2 negative cells. They further investigated the role of ACE2 shedding. A non-cleavable ACE2 mutant reduces syncytia, suggesting that ACE2 processing contributes to syncytia formation.

The article is well written, and the presented experiments are convincing and well controlled. The study relies on only one type of assay (transfection-based cell-cell fusion) and broad-spectrum metalloprotease inhibitors.

Main comments

1.     It would have been nice to confirm some of the results using virus/pseudotypes and knock-down of metalloproteases such as ADAM10 and/or ADAM17 to see if these enzymes are responsible for the observed phenotype.

2.     Fig.1. Why did the authors use 120 uM of camostat? This concentration is very high.

3.     Fig.2. It is important to control for the impact of the drugs on surface levels of ACE2 and S. This could be done by culturing the ACE2+ and S+ cells individually, thus avoiding fusion and staining for flow cytometry.

4.     Fig. 2 and 5. Images could be made bigger

5.     Lines 356-358 which justify the usage of HUH-7 cells, the cited reference (38) does not seem to be mentioning these cells.

6.     Fig 6. E-I. HEK293T Spike transfected supernatant is compared to media from untransfected cells. It is important to test media from cells transfected with a control plasmid.

7.     Line 582 The remaining spike protein-induced cell-cell fusion of the ACE2 L584A mutant was susceptible to hydroxamate-based metalloprotease inhibition (Figure 7F), suggesting that the ACE2 L584A mutant may still be cleaved by alternative metalloproteases.” I do not agree with this conclusion as: The WB in Fig. 7A shows that in absence of TMPRSS2, ACE2-L584A-CTF is not detected, suggesting that in these cells no metalloprotease cleaves ACE2-L584A. In fig. 7F, BATIM, MARIM and TAPI-1 all inhibit fusion of ACE2-L584A-CFT in the same conditions. This would suggest that the effect of the metalloproteases is not due to its action of ACE2 cleavage (as there is none detectable at baseline according to 7A) but most likely due to inhibition of Spike cleavage.

Author Response

Please see the attachment for response to comments and additional supporting data.

Reviewer 2 Report

In this manuscript, the authors found metalloproteases play a key role in SARS-CoV-2 spike protein-induced syncytiation in the absence of serine proteases. Some metalloprotease inhibitors could also inhibit the SARS-CoV-2 spike protein-mediated cell-cell fusion. Overall, the findings are novel and the study is thorough. The metalloprotease inhibitors might provide a novel strategy for the treatment of COVID-19 patients. There are though several points that should be addressed for the manuscript to warrant publication.

1.     Some membrane fusion inhibitors such as the EK1, a peptide targeting HR1 domain of S protein to block six-helix bundle formations, should be used as a positive control.

2.     The conclusion of “ACE2 ectodomain shedding is a potential contributor to SARS-CoV-2 syncytiation” seems not to be solid.

3.     Whether the combination of metalloprotease inhibitors and serine protease inhibitors would show a synergistic effect?

4.     The statistical methods should be indicated for every figure and need to be also described in the figure legends.

Author Response

Please see the attachment for response and additional supporting data
